Trends in fragmentation and connectivity of Paspalum quadrifarium grasslands in the Buenos Aires province, Argentina

Gandini Marcelo L. mgandini@faa.unicen.edu.ar 1 2
Lara Bruno D. 1 3
Moreno Laura B. 1
Cañibano Maria A. 1
Gandini Patricia A. 4 5
1 Facultad de Agronomía, Universidad Nacional del Centro de la Provincia de Buenos Aires , Azul , Buenos Aires , Argentina
2 CIC (Comisión de investigaciones Científicas de la provincia de Buenos Aires) , Azul , Buenos Aires , Argentina
3 CONICET (Comisión Nacional de investigaciones Científicas y Técnicas de Argentina) , Azul , Buenos Aires , Argentina
4 Unidad Académica Caleta Olivia, Universidad Nacional de la Patagonia Austral , Puerto Deseado , Santa Cruz , Argentina
5 CONICET (Comisión Nacional de investigaciones Científicas y Técnicas de Argentina) , Puerto Deseado , Santa Cruz , Argentina
Blackburn Jason
Electronic publication date: 2019 Feb 27
Publication date: 2019
Volume: 7
Electronic Location ID: e6450
Received 2018 Oct 8; Accepted 2019 Jan 14
Copyright: ©2019 Gandini et al.
Copyright year: 2019
Copyright holder: Gandini et al.
License: This is an open access article distributed under the terms of the Creative Commons Attribution License, which permits unrestricted use, distribution, reproduction and adaptation in any medium and for any purpose provided that it is properly attributed. For attribution, the original author(s), title, publication source (PeerJ) and either DOI or URL of the article must be cited.
License URL: https://creativecommons.org/licenses/by/4.0/

Keywords: Cattle grazing, Morphology, Guidos toolbox, Spatial pattern, Network analysis, Habitat loss, Land cover change, Community replacement

Funding: C.I.C (Commission of scientific research of the province of Buenos Aires) UNCPBA project Facultad de Agronomia de Azul CONICET, Argentina This work was supported by C.I.C (Commission of scientific research of the province of Buenos Aires), the UNCPBA project 03/A194 and Facultad de Agronomia de Azul. The participation of Dr. Bruno Lara was possible with a Doctorate Completion Grant from CONICET, Argentina. There was no additional external funding received for this study. The funders had no role in study design, data collection and analysis, decision to publish, or preparation of the manuscript.

==============================
Background

Despite its wide distribution worldwide, only 4.6% of temperate grasslands are included within systems of protected areas. In Argentina, this situation is even more alarming: only 1.05% is protected. The study area (central area of the southern Salado River basin) has a large extent of grasslands of Paspalum quadrifarium (Pq) which has been target since the middle of the last century of a variety of agricultural management practices including fire burning for cattle grazing.

Methods

Five binary images of presence-absence data of Pq from a 42-year range (1974–2016) derived from a land cover change study were used as base data. Morphological Spatial Pattern Analysis (MSPA), Morphological Change Detection (MCD) and Network Connectivity Analysis (NCA) were performed to the data using Guidos Toolbox (GTB) for the estimation of habitat and connectivity dynamics of the Pq patches (fragments).

Results

A loss of the coverage area and habitat nuclei of this grassland was observed during the study period, with some temporal oscillation but no recovery to initial states. Additional drastic reduction in connectivity was also evident in resulting maps. The number of large Pq grassland fragments (>50 ha) decreased at beginning of the study period. Also, fragmentation measured as number of components (patches) was higher at the end of the study period. The Pq pajonal nuclei had their minimum representativeness in 2000, and recovered slightly in area in 2011, but with a significant percentage increase of smaller patches (=islets) and linear elements as bridges and branches. Large corridors (mainly edge of roads) could be observed at the end of the study period, while the total connectivity of the landscape pattern drops continuously. Statistics of links shows mean values decreasing from 1974 to 2016. On the other hand, maximum values of links decreased up to 19% in 2011, and recovered to a 54% of their original value in 2016.

Discussion

Pq fragmentation and habitat reduction could have an impact on the ecosystem functioning and the mobility of some species of native fauna. The connecting elements of the landscape were maintained and/or recovered in percentage in 2011 and 2016. This fact, although favoring the dispersion of the present diversity in the habitat nuclei could cause degradation by an edge effect. Part of the area has the potential to be taken as an area of research and as an example of livestock management, since it is the one that would most preserve the biodiversity of the Pq environment. On the methodological side, the use of a proved tool as GTB is useful for monitoring dynamics of a grassland-habitat fragmentation.

Introduction

The loss of habitat and the fragmentation of ecosystems are one of the main threats to the conservation of biodiversity worldwide (Fahrig, 2003; Hobbs & Yates, 2003; Henle et al., 2004; Wilson et al., 2016; Woods et al., 2016). In Argentina, the conversion of natural ecosystems to agricultural lands has consequences such as the loss of habitat and biodiversity, the alteration of biotic interactions and biogeochemical processes (water cycles, carbon and nutrients), the reduction of the capacity to provide ecosystem services and the transformation of the landscape (Herrera, Texeira & Paruelo, 2013; Volante et al., 2012; Gandini, Lara & Scaramuzzino, 2014). In this way, and given the impact anthropogenic activities on natural systems (Vitousek et al., 1997), understanding if and how biodiversity recovers from disturbances is an important focus of ecology and conservation biology.

Despite its large distribution worldwide, only 4.6% of temperate grasslands are included within national protected area systems (Baldi et al., 2017). In Argentina, this situation is even more alarming since only 1.05% is protected (Bilenca & Miñarro, 2004). The underestimation of the productive value of these natural grasslands starts from the difficulty of objectively visualizing the goods and ecosystem services that they provide.

A characteristic type of grassland landscape in the Salado river basin is the “pajonal” of Paspalum quadrifarium also known as Paspaletum (Vervoorst, 1967). It represents one of the twelve plant communities identified in this area. It is a type of grassland characterized by a marked abundance of P. quadrifarium, a grass that can grow into dense tufts reaching 1 to 1.50 m high (Frangi, 1986), intermingled with various companion species in different proportions. Paspaletum is characterized by its distribution in a wide range of topographies (Lara & Gandini, 2013a) forming different vegetation units (Perelman, Burkart & León, 2003; Lara & Gandini, 2013b).

Paspaletum, like other pampa’s grasslands, has been under fire and grazing disturbance for a long time. Since the introduction of domestic livestock by European settlers and almost without interruption, this grassland has been managed by changing their coverage and land use in different ways (Foley et al., 2005; Vazquez, Zulaica & Requesens, 2012). Fire burning is currently used in the winter-spring period with the aim of increasing net productivity and thus livestock receptivity (Laterra, 2003). In this way, the interaction of fire with cattle grazing led to deep changes that can be seen across scales of analysis (Herrera et al., 2009; Lara & Gandini, 2011).

Ecologists distinguish between a particular disturbance event—like an individual storm or fire—and the disturbance regime that characterizes a landscape (Turner & Gardner, 2015; White & Jentsch, 2001). The current increase in disturbances due to agricultural and livestock pressure in the region (Cañibano, Gandini & Sacido, 2004; Vazquez, Zulaica & Requesens, 2012) is a discouraging scenario leading to the substitution of natural grass cover. However, Paspaletum (ecological community of Paja Colorada—Paspalum quadrifarium) remnant patches still persist in the centre of Buenos Aires province. These patches were maintained as pasture sites in good state of conservation (Herrera et al., 2009).

According to studies of “Fundación Vida Silvestre Argentina”—an environmental NGO—these sites were classified as Valuable Grassland Areas (PVAs), given their importance as a source of great native animal diversity and the numerous ecosystem services they provide (Bilenca & Miñarro, 2004).

Human encroachment on the environment through resource extraction and urban expansion have led to fragmentation (Maguire, Buddle & Bennett, 2016), with consequences for biodiversity (Chapin et al., 2000), ecosystem processes (Díaz & Cabido, 2001; Harrington et al., 2010) and the ecosystem services that they are supporting (Mitchell, Bennett & Gonzalez, 2014). Fahrig (2003) considered fragmentation as one of the most damaging threats to biodiversity conservation in recent times because the population viability in fragmented landscapes depends to a large extent on the structural and functional integrity of the landscape. In this context, it is necessary to carry out studies that allow analysing the trends of change at landscape scale in these habitat fragments, and their connections to implement management and conservation strategies in areas of high regional ecological importance and vulnerability.

The sustainable management of fragmented landscapes will depend on understanding the spatial ecology of the ecosystem services needed over the long-term (Maguire, Buddle & Bennett, 2016). In terms of the functional integrity of ecosystems, landscape connectivity is considered one of the key properties to maintain biodiversity. Landscape connectivity is defined as the degree to which landscape facilitates the movement of species and other ecological flows (Taylor et al., 1993). It is considered a key aspect to take into account for biodiversity conservation efforts around the world and one of the best responses to counteract the negative effects of habitat fragmentation and to facilitate species adaptation to changes in their natural habitats (Crooks & Sanjayan, 2006).

Improved landscape connectivity is becoming regarded as a viable management strategy to maintain biodiversity, ecosystem functions, and services (Ziter, Bennett & Gonzalez, 2013). In a part of the study area -the Salado River basin- the habitat fragmentation pattern has been reported to have increased considerably in the last 40 years (Lara & Gandini, 2014), thus becoming one of the major environmental problems in the basin.

Guidos Toolbox (GTB) is a free software available for analysing the structural and functional integrity of the landscape in terms of conservation. Particularly Morphological Spatial Pattern Analysis (MSPA) and Network Connectivity Analysis (NCA) can help researchers to build hypothesis about this issue. The generic setup of MSPA has been used to identify and map forest patterns, both structural (Vogt et al., 2006) and functional (Vogt et al., 2007), to identify key connectors for habitat suitability (Saura et al., 2011), for riparian corridor conservation studies (Clerici & Vogt, 2013), or evaluation of the US green infrastructure (Wickham et al., 2010) up to classifying zooplankton (Schmid et al., 2016). A unique feature of MSPA is the automatic detection of connecting pathways between core areas of image objects. Once found, the logical next point of interest is to rank those detected pathways with respect to the relative importance of each component, node and link in a given network. This task could be achieved by applying concepts and metrics of graph theory (Saura & Torné, 2009). Compared to typical Fragstat indices (Lara & Gandini, 2014), we consider that MSPA-exclusive information on core-areas, islets and connecting pathways could be the best way to show the changes along time. Complementarily, the direct relationship between MSPA and network theory presents NCA as the simplest way to explain the trends we search for. In addition, within GTB it is also possible to perform a calculation of the Equivalent Connected Area (ECA) as a summary of overall connectivity, because it has area units, it is easier to interpret and has a more usable range of variation (Saura et al., 2011).

In this work the MSPA and NCA in GTB were applied to a remote sensing classification of Landsat images in order to analyze the 42-year temporal change in the habitat fragments of Pq grasslands (Paspalum quadrifarium).

Materials & Methods

Study area

The study was carried out in the Flooding Pampa and Inland Pampa areas of the Salado River basin, in the Buenos Aires province (Argentina), covering mainly two different agroecological zones, the “Flooding and Inland Pampas” (Fig. 1). This square area contains most of the population of Pq and thus was considered as representative of the ecological community of interest (Gandini, Lara & Scaramuzzino, 2014) and their conservation status.

Figure 1 Study area and agroecological zones.

Study area and its location in the province of Buenos Aires, Argentina and South America.

This phase of the work closely follows the methods previously described in Lara & Gandini (2014), with the addition of some data sources not used or not available for that work.

Data acquisition, pre-processing and land cover classification

A series of five Landsat images (path 225, row 85) for the years 1974 (MSS sensor), 1988 (TM sensor), 2000 (ETM + sensor), 2011 (TM sensor) and 2016 (OLI sensor) was used. The digital numbers (DN) were converted to reflectance (except for the thermal band) according to Chander, Markham & Helder (2009), and the reflectance values were then adjusted for atmospheric scattering using the Improved Dark Object Subtraction method by Chavez Jr (1996).

In accordance with previous reports (Herrera et al., 2009; Lara & Gandini, 2013b), the initial land cover types used were: pajonal, short-grass matrix, pastures, crops and water bodies. To identify the land cover types, supervised classifications were employed using the maximum likelihood algorithm (Lu & Weng, 2007). The classifications were performed using all reflective Landsat bands; for 1974: MSS4, MSS5, MSS6 and MSS7; for 1988 and 2011: TM1, TM2, TM3, TM4, TM5 and TM7; for 2000: ETM + 1, ETM + 2, ETM + 3, ETM + 4, ETM + 5 and ETM + 7, and for 2016, Bands 2, 3, 4, 5, 6 of OLI Sensor. The thermal bands of platforms were discarded.

For 1988, the training sites were located by visual interpretation on 44 aerial infra-red photographs (scale 1: 20,000) taken in 1988 summer, following criteria such as texture, shape and colour (Chuvieco, 2010). For 2011 and 2016, the control points were selected using a global positioning system (GPS) in the field within relatively homogeneous areas. For 1974 and 2000, the training sites were selected by visual analysis following the medium spectral signature for each land cover type and using areas with similar spectral characteristics—over land cover remained unchanged—(Chuvieco, 2010; Schulz et al., 2010).

Classification results were filtered using a 7 × 7 median filter to remove isolated pixels. Later, the MSS classification (1974) was re-sampled to a 30 × 30 m pixel size to allow multi-temporal comparison with the rest of the series.

The accuracy of the classification maps was assessed with the use of quantity disagreement and allocation disagreement (Pontius & Millones, 2011). These indices are more useful and simpler than standard Kappa (Congalton, 1991) and allowed us to focus on two components of disagreement between maps and reference points in terms of the quantity and spatial allocation of the land cover types. Quantity disagreements, and not allocation disagreement, were particularly taken into consideration to assess classification quality since the main aim of this work was focused on regional changes and not pixel-to-pixel changes (Keller & Smith, 2014).

MSPA and NCA

Pajonal fragments were identified based on these classifications, and binary Pajonal presence-absence maps were created. These binary maps of 1974, 1988, 2000, 2011 and 2016 were analyzed using MSPA within GTB (Vogt & Riitters, 2017).

MSPA Parameters were chosen taking into account field measurements where it was shown that patches of approximately 1ha (three Landsat pixels) may contain maximum diversity of plant species, and that it also depends on the distance and connectivity between the patches. In this way, an edge width of three pixels and the standard foreground connectivity of 8 was chosen. Besides this, we divided core-areas into small/medium/large using the default settings of 1,000 and 4,600 pixels in order to consider the size of cattle ranches and the type of megafauna present in the study area, and also to use this distinction to visually highlight the changes over time.

At the same time, a comparative analysis of trends in landscape connectivity was carried out, evaluating the variations in the networks and connectivity of the patches of pajonal. The analysis was performed considering each set of connected patches as a single landscape element. In this way the size of the elements varied as the patches were fragmented and disconnected by the effect of the livestock management (fire and grazing disturbance).

ECA was calculated via GTB. ECA measures the overall structural connectivity between the set of network components (Core + bridges but excluding all the other MSPA classes) accounting for their area and the number of nodes and links within each component.

Because of this definition, ECA provides us a measure which is not species-specific and as such an independent overall measure of connectivity (Saura et al., 2011).

On the other hand, spatio-temporal changes were analyzed by Morphological Change Detection (MCD) within GTB with a focus on neglecting spurious changes and providing essential change in areas of loss and gain only (Seebach et al., 2013).

Results

The supplemental material provides a complete set of base images, and resulting images of Classification (Pq mask), MSPA and connectivity for the 5 analyzed years. To simplify interpretation, only the start and end years of the observation time period (1974–2016) are shown in the main body of the text.

Image classification of land cover

Classification error issues are shown in Table 1. Good thematic accuracies for Pq were obtained, and the separation of overall disagreement into two components was used to learn about sources of error and give guidance on how to improve each classification (Pontius & Millones, 2011).

Table 1 Classification quality of Pq cover estimation, and numerical results of the morphological and connectivity analysis.

Most significant MSPA results (as percentages over the total) and connectivity analysis: Number of components, maximum, mean and median of connectors/component; and relative ECA.

Analysis	Parm/Form/year	1974	1988	2000	2011	2016	
Class. Disagreement % (PJ)	Quantity	1	5	5	4	2	
Allocation	10	18	12	9	12	
MSPA	Core (small)	4.04	3.12	3.68	3.17	3.46	
Core (medium)	4.43	2.08	3.12	2.85	2.17	
Core (large)	16.95	4.58	6.06	3.73	3.71	
Islet	0.85	2.53	1.11	1.46	1.75	
Perforation	2.97	0.73	0.60	0.30	0.57	
Edge	17.06	9.87	11.13	8.61	10.72	
Loop	0.19	0.40	0.27	0.14	0.20	
Bridge	0.46	2.24	1.11	0.73	0.77	
Branch	1.91	3.24	2.35	2.13	2.31	
Foreground Area	# pixels	7,878,004	4,641,009	4,745,490	3,728,019	4,135,826	
Conectivity (links)	Mean of links	1153	379	287	167	49	
Median of links	55	49	41	21	43	
Max links	80,557	28,468	16,738	15,040	43,912	
ECA (pixels)	945,277	894,122	1,041,192	236,552	374,184	
relative ECA %	28	57	50	15	25	

Fragmentation, habitat loss and trends in connectivity

Table 1 summarizes the results of MSPA and Connectivity numerically. The small cores had a rapid fall of around 25% between 1974 and 1988, and then kept oscillating between values that never reached the initial stage. A similar behaviour was found for medium cores, which dropped by approximately 50% in the same time frame. The main change was found for large cores with a decrease of more than 60% in 1988, followed by a transient slight increase in 2000, and continued with the decreasing trends in 2011 and 2016.

The edges tend to decay oscillating until reaching a minimum value of 8.61 in 2011 and then recover slightly in 2016. Islets and perforations show a similar trend like cores.

Regarding the “linear” elements (loops, bridges and branches) a tendency to prevail in time is observed although the bridges are lost more quickly reaching low values from 2000.

The foreground area, reached a minimum value in 2011, recovering slightly in 2016 (Table 1, see also Figs. 2 and 3), although not achieving the previous values of 1974, 1988 or 2000.

Figure 2 MSPA results for 1974.

Results showing start (1974) of Pq grassland status. In different colors, core areas and another morphological elements obtained by MSPA (see references) Background areas in grey.

Figure 3 MSPA results for 2016.

Results showing end (2016) of Pq grassland status. In different colors, core areas and another morphological elements obtained by MSPA. Background areas in grey (not discussed).

Statistics of links showed mean values decreasing from 1974 to 2016, and maximum values recovering to a 50% of their original value. In terms of connectivity, the average number of links declined rapidly between 1974 and 1988, tending to decrease during the entire study period, although the percentage in relative ECA increased at a high rate and there was a minimum in 2011, indicating the dynamics of the patches.

Fragmentation measured as number of components (patches) resulting from habitat pressure was higher in 2011 and 2016 (Table 1). This observation agrees with MSPA data, in which the pajonal nuclei had their minimum representativeness in 2000, and recovered by 2011, with a significant percentage increase in the small patches (=islets). While edges and loops remained relatively stable, the linear elements (bridges and branches) increased their representativeness (Table 1, MSPA; see also Figs. 2 and 3). In 2016, larger corridors (mainly edges of roads) could be observed while the total connectivity of the landscape pattern remains low (Table 1; connectivity and Figs. 4 and 5) in terms of mean and median connectivity after 1988 and percentage of relative ECA after 2011.

Figure 4 Network connectivity results for start of period of study (1974).

Objects belonging to the same network are represented with the same color.

Figure 5 Network connectivity results for end of period of study (2016).

Objects belonging to the same network are represented with the same color. A comparison with Fig. 4 can spatially indicate the relevance of the change in connectivity.

Temporal variation of relative ECA showed the dynamics of the fragmentation process in the study area: from an initial value of 28%, almost doubled in only 14 years (1988). In successive years, the connectivity and ECA decreases to a minimum in 2011, followed by a slight increase in 2016. A visual comparison between connectivity maps (Figs. 4 and 5), showed spatially the relevance of the change in connectivity and network size.

Figure 6 shows the result of MCD applied between 1974 and 2016. A foreground loss of 43% is explained mainly as an internal foreground loss (elasticity 1.10575). In general, it could be observed a high Pq loss area and some gain areas in the west side of the map. Only a zone in the south-west of the study area remains less changed in the foreground.

Figure 6 Morphological Change Detection between 1974 and 2016.

Results of applying MCD to data from start and end of study period. Pixels in red indicate loss of foreground area, and green indicate gains. A foreground loss of 43% is explained mainly as internal foreground loss (elasticity 1.10575) Dark grey indicate unchanged foreground area.

Discussion

This work showed that the habitat patches of Pq have undergone deep transformations since 1974 in agreement with the findings of Lara & Gandini (2014). They reported that the original landscape matrix of Pq was replaced by other subordinate communities, e.g., Short-Grasses Matrix.

The variations in the size and connectivity of the patches of Pq were significant in the study period (1974–2016). The major effects were the habitat fragmentation and patch number and size reduction leading to connectivity loss. In terms of fragmentation, there was a true fragmentation, and also a slight habitat loss, determined by: (a) a high trend to lower the patch size and the connectivity and (b) an increase of linear elements at the end of the study period. Although the connecting elements of the landscape had maintained and/or recovered in percentage, favoring the dispersion of the present diversity in the habitat nuclei and preserving patches as breakup centers, this fact could be related with a degradation by edge effect.

Slight structural recovery measured by ECA% and in terms of area could be seen at the end of the study period mainly for the maintenance of linear structures such as bridges. Similar to fragmentation, connectivity is not a good or bad issue, and it strongly depends on the selected species we analyse but a negative trend is pointing at least some habitat quality loss for a species.

Statistics of links as mean, median and maximum, shows also the trends of connectivity and habitat losses. Means decreased continuously, and median values increase in 2016 along with the maximum number of links indicating a possible partial recovery.

The effect on diversity could be more important when considering that habitat patches form a “network” and that they are not connected to each other (Maguire, Buddle & Bennett, 2016). This loss of connectivity is clearly evidenced comparing Figs. 4 vs 5.

Although the connectivity (median values) of 1988 is maintained with respect to 1974, relative ECA increases to double. This would indicate that the dynamics of fragmentation occurred rapidly, starting with a decrease in the size of the patches, followed by a loss of connectivity. The ECA as calculated via GTB looks at the overall structural connectivity between a set of network components (Core + bridges of Pq) but excluding all the other MSPA classes, accounting for their area, number of nodes and links within each component. ECA is defined as the size that a single patch of habitat (maximally connected) should have in order to provide the same value of IIC (integral index of connectivity) or PC (Probability of Connectivity) than the actual habitat pattern in the landscape (Saura et al., 2011). With this definition, ECA provides an non species-specific measure and as such an independent overall measure of connectivity.

Within the framework of environmental conservation, monitoring these environments and conducting further research on their impact on biodiversity are deemed necessary. The methodological approach used in this work, like that employed by Wickham et al. (2010), was used to identify and classify the morphological types of fragmentation, focusing on the dynamics of habitat availability and also to recognize the temporal variation of the landscape connective elements.

This research opens up the need to face a short-term research about the minimum size of pajonal fragments that preserve different ecosystem services, maintaining their biodiversity components and structure in an acceptable status. Having this information, it is possible to propose sustainable management plans for the livestock agro-ecosystems of the region.

The results obtained with MCD also show that the community of Pq (managed as forage source in this case) tends to change spatially in a very dynamic way as shown clearly in Fig. 6. Zones without change are very scarce during the study. Only a portion of the study area in the south–west remains substantially unchanged in the foreground. This area has the potential to be taken as an area of research and also as example of livestock management, since it is the one that would most preserve the biodiversity of the Pq environment.

On the methodological side, the generic design of the fragmentation assessment scheme employed here for grasslands could be applied equally to assess and monitor dynamics of any other land cover type, i.e., fragmentation of forests or agricultural land.

Supplemental Information

Supplemental Information 1 Set of binary presence-absence files used as base data for the analysis

Each file represents a year of fragmentation and connectivity analysis.

Click here for additional data file.

We wish to acknowledge Professor Mariana Oyarzabal, Luciano Gandini and Maximiliano Gandini for their contribution in english grammar revision. We also want to thank Dr. Vilma Manfreda for her critical reading of the different versions of the manuscript. A special thanks to Dr. Peter Vogt, who has helped us increase the quality of this manuscript.

Additional Information and Declarations

Competing Interests

Author Contributions

Data Availability

Patricia Gandini is an Academic Editor for PeerJ.

Marcelo L. Gandini conceived and designed the experiments, performed the experiments, analyzed the data, prepared figures and/or tables, authored or reviewed drafts of the paper, approved the final draft.

Bruno D. Lara conceived and designed the experiments, performed the experiments, analyzed the data, authored or reviewed drafts of the paper, approved the final draft.

Laura B. Moreno performed the experiments, contributed reagents/materials/analysis tools, authored or reviewed drafts of the paper, approved the final draft.

Maria A. Cañibano performed the experiments, authored or reviewed drafts of the paper, approved the final draft.

Patricia A. Gandini performed the experiments, analyzed the data, authored or reviewed drafts of the paper, approved the final draft.

The following information was supplied regarding data availability:

The raw data are available in a Supplemental File.

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
