# Peer review of "Trends in fragmentation and connectivity of Paspalum quadrifarium grasslands in the Buenos Aires province, Argentina"

_PeerJ, doi:10.7717/peerj.6450_

## Round 0.1 · original submission · Major Revisions

Thank you for your interesting paper. Each of two reviewers have provided substantial input into reviewing and providing important comments for your consideration. Please note, while Reviewer 2 suggested a rejection, the reviewer did provide detailed comments as to why and I feel those comments result in a Major Revision. Reviewer 1 provided significant detail to revise the paper and working through both it is likely the paper is publishable.

Both reviewers note, and I concur, that the manuscript requires significant changes to the methods and results section. Following Reviewer 2, please be careful to separate results and discussion. Prior to reconsideration for publication, the English will need to be improved substantially.

Please work carefully through these revisions and resubmit.

·

Basic reporting

The manuscript describes trends in fragmentation and connectivity of grasslands in the Buenos Aires province, Argentina using 5 satellite images from 1974 to 2016. The introduction is well developed and sets the scene with adequate references. Figures are relevant but could be improved with more detailed captions. I appreciate providing the raw data to the readers so they can replicate the steps described in the paper.

Experimental design

The design setup is within the scope of the journal, research questions are well defined and relevant. The method section provides many details on the data preparation but very little detail on the actual processing of the data, esp. for connectivity. Since the topic of the paper is about trend analysis, the reader expects to find sufficient details describing how exactly that trend analysis was conducted but right now there is not much information about this topic.
Similar to Connectivity, the notion of fragmentation is a very wide field and the authors should explain why they have chosen MSPA for their analysis, or which features of MSPA are considered to provide more appropriate information compared to typical Fragstats indices, as used in the Lara & Gandini paper 2014 (in the supplemental data). For example, I could imagine that the MSPA-exclusive information on core-areas, islets and connecting pathways and their direct relation to network theory analysis used to calculate the degree of network connectivity, was a decisive argument.

Validity of the findings

The findings are sound, relevant and to my knowledge new in the field. However, I think the paper would benefit from additional information and/or reasoning on the chosen settings for scale, MSPA-settings, and connectivity assessment.

a) In the introduction you mention the importance of spatial scale, which is good. But where and how is this topic addressed in your analysis?

b) MSPA analysis:
- you seem to have used an older GTB version of the 2.6 series. Please note that a small bug in the MSPA statistics calculation has been fixed and more recent versions also provide information on different type of perforations within Core area, a topic which is of interest when talking about fragmentation. You could reprocess the 5 images (only takes some seconds) to get updated and enhanced MSPA information for your 5 images.
- you use EdgeWidth=3, why did you opt for a 3-pixel edge width, because that corresponds to roughly 100m (90m to be precise)?
- you selected to divide core-areas into small/medium/large using the default settings of 1000 and 4600 pixels. Can you explain why you choose to show the core-area distinction and why you used the two thresholds and not others, for example 1000 and 10000? Maybe you use the distinction to visually highlight the locations of large/medium/small core areas and/or analyse how these types have changed over time?

c) connectivity analysis
You may add some comments on connectivity. Similar to fragmentation, connectivity is not good or bad, and it strongly depends on the selected species. The same habitat can be well connected for one species but not connected at all for another species. The ECA as calculated via GTB looks at the overall structural connectivity between the set of network components (Core + bridges but excluding all the other MSPA classes) accounting for their area and the number of nodes and links within each component. Because of this definition, ECA provides a measure which is not species-specific and as such an independent overall measure of connectivity.


The supplemental raw data supplied is somewhat confusing:
- Binary images are provided for 1974, 1988, 2000, 2011, 2016: all images should be provided in byte format instead of integer to allow reproducing the analysis within GTB.
Looking in the subfolder "frequency" we find that the actual image for 1974 is actually from 1975 with the name "pajonal1975_r", which must be in byte format. Maybe better to provide the actual data you used in your analysis instead of the integer formatted image "1974.tif"
- The information in the subfolder "GToolbox" is superfluous as it is identical to the one in the subfolder "frequency"
- the purpose of the paper in the subfolder "Paper" is unclear to me, or please add an explanation why it is needed as supplemental material.

Additional comments

The paper describes dynamics in Pq grassland fragmentation and connectivity. But the title puts a strong focus on MSPA without mentioning fragmentation and connectivity. If the importance is MSPA then maybe there should be a dedicated section on MSPA, highlighting why this and not any other methodology was chosen. Or the maybe the title could be changed to better reflect the actual focus of the paper, something like:
Trends in fragmentation and connectivity of Paspalum quadrifarium grasslands in the Buenos Aires province, Argentina.

Specific remarks at line number:
25: correct: Guidos Toolbox -> GuidosToolbox (GTB), then any later reference can be: GTB

27: We found a loss in total coverage area, number of habitat nuclei and a reduction in connectivity. The number of large Pq....

38: GTB is not a forest fragmentation software but instead a generic image analysis toolbox, which can be used for various analysis schemes, fragmentation is just one of them. Either delete the last sentence at line 38 (because you do not talk about forest fragmentation in the paper) or you could say:
On the methodological side, the generic design of the fragmentation assessment scheme employed here for grasslands could be applied equally to assess and monitor dynamics of any other land cover type, i.e., fragmentation of forests or agricultural land.

100: add a reference for the statement "the numerous ES they provide".

115-137: Morphological Spatial Pattern Analysis (MSPA, Soille & Vogt, 2008) is a customized sequence of mathematical morphological operators targeted at the description of the geometry and connectivity of image objects. Starting with a binary representation, MSPA will conduct a segmentation of the image objects into various morphological feature classes, including core, islet, loop, bridge, perforation, edge, and branch. A summary description and application examples can be found on the MSPA website.
[MSPA website: http://forest.jrc.ec.europa.eu/download/software/guidos/mspa/
Below the animated image on that website you can find links to papers where MSPA was used]
If you want to go into details...
The generic setup of MSPA has been used to identify and map forest patterns, both, structural (Vogt et al 2007a, 2007b) and functional (Vogt et al., 2009), identify key connectors for habitat suitability (Saura et al., 2011), riparian corridor conservation studies (Clerici and Vogt, 2013), or evaluation of the US green infrastructure (Wickham et al., 2010) up to classifying zooplankton (Schmid et al, 2016).

142: maybe delete the sentence "In this way,..." and just keep the last sentence of the introduction

185: .. using MSPA within GTB (Vogt and Riitters, 2017).

187 - 191: provide details how you did the connectivity analysis. Explain the reasons why you think the selected method is appropriate for your project.

192 - 195: provide details how you calculated ECA, with GTB or with Conefor, or how? Explain the reasons why you consider ECA to be a meaningful measure for your project.

192: In addition, we calculated the Equivalent Connected Area (ECA, Saura et al 2011), defined as ...

201: To simplify interpretation we only show the start and endpoint of the observed period in the main body of the text, all results can be found in the supplemental material.

212: "A drastic reduction in connectivity is also evident." I can not see that in Figure 2, maybe you refer to the table 1?
A suggestion: In addition to Figure 2 it might be interesting to show the area change (morphological change detection) from 1974 to 2016, showing where, and how much, grassland cover was lost/gained and where it remained unchanged. This can be done quickly in GTB (File -> Change -> Morph.Change), please find the result here:
https://www.dropbox.com/s/0837puze26r9uh7/morphdiff.tif?dl=0
With these statistics:
MCD (A->B) FG: -43.0117, FGi: -47.5600, Elasticity: 1.10575
Gain [pixels]: 2113195
Loss [pixels]: 5147462
Indicating that the loss has predominantly taken place within existing grasslands and overall 43% of grasslands were lost since 1974. This result is more sound than a visual comparison of the two images in Figure 2.

213: where can I find the number of large fragments (> 50 ha)?

219: Table 1 does not suggest that Edges remain stable and what are "curls"?

226: %ECA did not drop but has increased after 2011.

229: why is the loss of connectivity evident in Figure 3? Compared to 1974, in 2016 the total number of components has increased and the total area has decreased => more fragmented => less connected.


References:

376: Vogt P, Riitters KH (2017) GuidosToolbox: universal digital image object analysis. European Journal of Remote Sensing, 50, 1, pp. 352-361, DOI: 10.1080/22797254.2017.1330650

Vogt P, Ferrari JR, Lookingbill TR, Gardner RH, Riitters KH, Ostapowicz K, 2009. Mapping Functional Connectivity. Ecological Indicators 9: 64-71. DOI: :10.1016/j.ecolind.2008.01.011


Figure 2: replace "References" with "MSPA classes", add a bar outlining the actual area of the image shown in km in x/y direction. This figure shows the study area as a quadratic image but the actual image in the supplemental data clearly is rectangular having the dimension 5168 x 3120 pixels.

·

Basic reporting

- English writing could be improved throughout. Specifically, many paragraphs include many fragments of ideas, jumping from one theme to another. Keep paragraphs focused on a single idea.

- Length of the introduction was far too long compared to overall length of manuscript. Specifically, > 50% (147/257 lines of text) of the manuscript text was in the introduction. The manuscript needs to be rebalanced with a tighter introduction and more detail in the methods & results and a more thorough discussion.

Experimental design

- The image classification methods employed seem standard, but these could be discussed in much more detail.

- There was inadequate detail in the methods about the MSPA method, and the subsequent connectivity analysis. Some of this appears in the introduction, this could be moved and improved.

- What is the relevance of the perfectly square study area?

Validity of the findings

- The classification results seem appropriate for the study.

- The results need to be elaborated upon in more detail in the text. Specifically, the numerical results in the table need improved reporting. It would be beneficial to separate the results from the discussion.

- The data was sourced over multiple years, then only results for 1974 and 2016 are shown. Why not all years?

- What can be gleaned from these maps exactly, it is difficult to tell. Needs more explanation about what is being seen. MSPA is a useful method but needs to be contextualized and analysed further, simply producing these maps without providing a more detailed summary analysis (e.g., what do the numbers in table 1 mean?) is not really useful.

Additional comments

- The manuscript does not provide enough detail about the methods or explanation of the results.

- The discussion is intertwined with the results and needs to be separated. Specifically, I would like more here about how the results compare to previous studies in the area, or similar studies elsewhere.

- Study area map could be improved greatly. Specifically, I’m not sure what projection was used but this seems like an odd choice for Argentina, also, Argentina is not an island. The coloring of the agroecological zones could be better suited to assist the reader in understanding what all these zones represent. More info on the study area is warranted in the text for those not familiar with the region.

---

## Round 0.2 · Minor Revisions

Your manuscript is as good as Accepted - I thank you for a full faith effort to revise the paper and integrate the suggestions of the reviewers.

I am returning it to you so you can make the short list of changes noted in the review of the revision. Please make these changes to complete this process.

·

Basic reporting

The manuscript describes trends in fragmentation and connectivity of grasslands in the Buenos Aires province, Argentina using 5 satellite images from 1974 to 2016. The introduction is well developed and sets the scene with adequate references. The revised Figures are relevant. I appreciate providing the raw data to the readers so they can replicate the steps described in the paper.

Experimental design

The design setup is within the scope of the journal, research questions are well defined and relevant. Methods are described sufficiently and reviewer comments were integrated.

Validity of the findings

The findings in the revised manuscript are sound and relevant.

Additional comments

Thank you for ingesting the comments on the original manuscript. The revised version reads much better. However, I still think the following minor changes should be inmplemented:
Line 55: In this way, and given the impact of

Line 116: .. (MSPA) and Network ...

Line 125: replace Vogt et al, 2017a with Saura & Torné (2009)
Saura, S. and Torné, J. (2009). Conefor Sensinode 2.2: a software package for quantifying the importance of habitat patches for landscape connectivity. Environmental Modelling & Software 24: 135-139.
DOI: 10.1016/j.envsoft.2008.05.005


Line 181: and the standard foreground connectivity of 8 was chosen.

Line 190: ECA was calculated via GTB. ECA measures the overall ...

Line 197: replace Vogt & Riitters, 2017 with Seebach et al. 2013
Seebach, L., Strobl, P., Vogt, P., Mehl, W., San-Miguel-Ayanz, J. (2013) Enhancing post-classification change detection through morphological post- processing – a sensitivity analysis. International Journal of Remote Sensing, 34 (20), 7145–7162. DOI: 10.1080/01431161.2013.815382

Line 199: The supplemental material provides a complete set of ...

Line 200: To simplify interpretation, only the start and end years of the observation time period (1974-2016) are shown in the main body of the text.


Line 209: ...around 25% between 1974 and 1988, and then kept oscillating between values that never reached the initial stage. A similar behaviour was found for medium cores, which dropped by approximately 50% in the same time frame. The main change was found for large cores with a decrease of more than 60% in 1988, followed by ....

Line 216: do you mean: Islets and perforation show a similar trend like cores. ??

Line 225: although the percentage of relative ECA increased at a high rate ...

Line 234: and percentage of relative ECA after 2011.

Line 236: of relative ECA showed...

Line 237: .. of 28%, almost

In general: the term "relative ECA" already implies relative values (percentages) so there is no need to write "relative ECA%", just replace it by "relative ECA"

Line 276: With this definition, ECA provides an non species-specific measure...

Figure 2/3: the image shows 3 different types of Core area (dark/medium/light green - small, medium, large Core areas) but the legend does not reflect this distinction.
Morphological elements -> morphological pattern classes.

Figure 6, legend: .... 1.10575). Dark grey indicate zones with unchanged

---

## Round 0.3 · accepted · Accept

Thank you for the attention to detail to make these minor revisions. The work was greatly improved in the first revision and this second revision completes the minor requests for clarification. I look forward to seeing your full paper published soon. Congratulations and nice work.

#